# Role of Microbiota-Gut-Brain Axis in Regulating Dopaminergic Signaling

**DOI:** 10.3390/biomedicines10020436

**Published:** 2022-02-13

**Authors:** Sevag Hamamah, Armin Aghazarian, Anthony Nazaryan, Andras Hajnal, Mihai Covasa

**Affiliations:** 1Department of Basic Medical Sciences, College of Osteopathic Medicine, Western University of Health Sciences, Pomona, CA 91766, USA; sevag.hamamah@westernu.edu (S.H.); armin.aghazarian@westernu.edu (A.A.); anthony.nazaryan@westernu.edu (A.N.); 2Department of Neural and Behavioral Sciences, College of Medicine, The Pennsylvania State University, Hershey, PA 17033, USA; axh40@psu.edu; 3Department of Biomedical Sciences, College of Medicine and Biological Science, University of Suceava, 7200229 Suceava, Romania

**Keywords:** dopamine, *Prevotella*, *Bacteroides Lactobacillus*, *Bifidobacterium Clostridium*, *Enterococcus*, *Ruminococcus*, Parkinson’s disease

## Abstract

Dopamine is a neurotransmitter that plays a critical role both peripherally and centrally in vital functions such as cognition, reward, satiety, voluntary motor movements, pleasure, and motivation. Optimal dopamine bioavailability is essential for normal brain functioning and protection against the development of neurological diseases. Emerging evidence shows that gut microbiota have significant roles in maintaining adequate concentrations of dopamine via intricate, bidirectional communication known as the microbiota-gut-brain axis. The vagus nerve, immune system, hypothalamus–pituitary–adrenal axis, and microbial metabolites serve as important mediators of the reciprocal microbiota-gut-brain signaling. Furthermore, gut microbiota contain intrinsic enzymatic activity that is highly involved in dopamine metabolism, facilitating dopamine synthesis as well as its metabolite breakdown. This review examines the relationship between key genera of gut microbiota such as *Prevotella, Bacteroides, Lactobacillus, Bifidobacterium, Clostridium,*
*Enterococcus,* and *Ruminococcus* and their effects on dopamine. The effects of gut dysbiosis on dopamine bioavailability and the subsequent impact on dopamine-related pathological conditions such as Parkinson’s disease are also discussed. Understanding the role of gut microbiota in modulating dopamine activity and bioavailability both in the periphery and in the central nervous system can help identify new therapeutic targets as well as optimize available methods to prevent, delay, or restore dopaminergic deficits in neurologic and metabolic disorders.

## 1. Introduction

The human gastrointestinal (GI) tract is host to trillions of commensal microbes, which play an integral role in the maintenance of the host immune and homeostatic mechanisms. Collectively, the genomic composition of the microbes contained within the human gut is referred to as the microbiome, while the collection of bacteria, archaea, and eukarya that reside there is termed microbiota [1,2]. Of the six main phyla of bacteria predominating the mammalian gut, Bacteroidetes and Firmicutes comprise approximately 90% of the total microbial population [3]. For the past decade, an abundance of studies have examined the symbiotic relationship between the mammalian host and the gut microbiota as it pertains to microbiota’s role in immunity, nutrient and drug metabolism, bone formation, growth maintenance of mucosal barriers, and protection against pathogens, to name a few [4,5]. More specifically, gut microbes have been implicated in endocrine functions via improvements in insulin sensitivity [6], reduction of intestinal inflammation [7], and facilitation of digestion through production of short-chain fatty acids (SCFAs) from complex carbohydrates [8,9,10]. The importance of the gut microbiota has been further supported by studies primarily demonstrating associations between the degree of microbial abundance, richness, and diversity, and some of the most prevalent diseases, such as obesity [11], type 2 diabetes mellitus, and metabolic syndrome [12,13], heart failure [14], and even cancers [15]. Therefore, it is not surprising that significant research efforts have been directed toward better understanding the effects that various microbes may have on both normal and pathophysiological processes in humans and their potential therapeutic implications. The regulation of these intricate processes involves a complex and dynamic bidirectional communication between the gut and the brain via the so-called gut–brain axis [16,17,18]. For example, several studies have shown that changes in the gut microbiome composition through manipulation techniques such as probiotic administration or fecal microbiota transplant can cause changes in brain activity and cognitive behavior by modulating neurotransmitter activity [19,20]. Dopamine, in particular, has drawn significant interest due to its contribution to pathological conditions in both the GI tract and the central nervous system [21,22,23]. The dopaminergic pathway contains neurons that reside in the ventral tegmental area (VTA) and substantia nigra pars compacta (SNpc) of the midbrain, which project to the nucleus accumbens via the mesocortical or mesolimbic pathways and dorsal striatum via the nigrostriatal pathway, respectively [24]. The dopamine transporter (DAT) and dopamine receptors (D1–D5) are essential in carrying out the numerous functions of dopamine and have roles that include, but are not limited to, cognition, reward, satiety, voluntary motor movements, intestinal motility, and secretions [25,26,27]. For example, D3 receptor deletion increases GABA-induced inhibition in the nucleus accumbens, subsequently reducing reward-mediated voluntary alcohol intake [28]. It has also been shown that increased D2 receptor postsynaptic signaling interferes with the firing patterns of dopaminergic neurons in the VTA [29]. In turn, this reduced cortical dopaminergic bioavailability leads to cognitive deficits. Additionally, rescuing deficiencies in fear extinction has been associated with enhanced gene expression of D1/D2 receptors in the prefrontal cortex (PFC), indicating the role of dopamine receptors in the onset of stress-related pathologies [30,31]. Further, changes in DAT and D1/D2 receptor expression in the striatum have been associated with alterations in the gut microbiome [20,32,33]. More specifically, certain microbes belonging to genera such as *Prevotella, Bacteroides, Lactobacillus, Bifidobacterium, Clostridium, Enterococcus,* and *Ruminococcus* have been shown to modulate receptors, transporters, and specific targets of the dopaminergic pathway, either in a positive or negative manner [20,34,35,36]. Although several studies have demonstrated important links between microbes and dopaminergic pathways both within the central nervous system (CNS) and peripherally in the GI system that result in behavioral changes [37], the underlying mechanisms are yet to be fully elucidated.

In this review, we present emerging evidence demonstrating the relationship between gut microbiota and dopaminergic pathways that impact behavior. As such, we briefly describe the mechanisms of reciprocal communication between peripheral and central nervous systems via the gut–brain axis and the role of gut microbiota in this process. Then, we present evidence showing the ability of gut microbes belonging to *Prevotella, Bacteroides, Lactobacillus, Bifidobacterium, Clostridium, Enterococcus,* and *Ruminococcus* to modulate dopaminergic activity and how subsequent alterations contribute to the pathogenesis of dopamine-related disorders. Overall, this review highlights the link between the gut microbiota and the dopaminergic system by presenting current evidence while identifying gaps and proposing future research directions to elucidate the potential contribution of gut microbiota to neurodegenerative disorders affecting dopaminergic neurons.

## 2. Microbiota-Gut-Brain Axis

The influence of gut microbiota is not limited to the gut but is a major player in the bidirectional communication between the gut and the brain. For example, disturbance of the healthy intestinal flora has been associated with numerous pathological conditions including neurological disorders such as Parkinson’s disease, ADHD, depression, anxiety, and autism [38]. It is not clear yet whether changes in the gut microbiota are the results of faulty brain signaling or whether they can actually drive brain disorders. It is known, however, that several intestinal microbes are involved in neurotransmitter synthesis such as glutamate, serotonin, noradrenaline, dopamine, and γ-aminobutyric acid (GABA) and their functions and bioavailability both in the CNS and in the periphery [39,40]. Changes in some of these neurotransmitters such as decreased dopamine concentrations have been linked with the etiology of neurodegenerative disorders. These neurotransmitters activate nerve ganglia in the myenteric and submucosal plexuses of the enteric nervous system (ENS) and are important mediators in the interface between the intestinal tract and the brain through the gut–brain axis. For example, enteroendocrine cells activate vagal afferents by using glutamate and serotonin as neurotransmitters, transducing luminal stimuli through the vagus nerve to the CNS [41].

In addition to an indirect effect mediated through the vagus nerve, microbes can interact directly with the CNS by direct actions in the GI tract, for example, via serotonin secretion into the lamina propria [42] or translocation of metabolites and endotoxins from the intestinal lumen to the main circulation [43,44,45]. Vagal afferents are also activated by numerous other signals, including hormones, nutrients, and peptides that are produced by intestinal microbes. Therefore, alterations in gut microbial species and the metabolites they produce can modulate afferent vagal nerve activity by multiple mechanisms. Indeed, SCFA-producing microbes were shown to activate and directly act on intestinal vagal terminals [46], and the introduction of *Lactobacillus rhamnosus* into the small intestine increased vagal afferent firing frequency [47]. Similarly, *Lactobacillus johnsonii* was shown to induce activation of vagal sensory neurons innervating the GI tract [48] and ablation of the vagus blocked anxiolytic and gene expression effects of *Lactobacillus rhamnosus* [49]. Together, these findings provide strong support for the role of the vagus nerve in mediating the effects of gut microbes on brain structures receiving direct viscero-sensory input. It should be noted, however, that the reported effects of gut microbes on the brain or whether the vagus nerve is involved in this effect is strain-dependent. For example, unlike *Lactobacillus rhamnosus (JB-1),* the effects of *Lactobacillus salivarius* on the enteric nervous system was not neurally dependent [50]. It is not clear, however, whether the vagus nerve is directly activated by microbes or rather indirectly via its soluble metabolites. Nonetheless, compelling evidence exists demonstrating how specific microbes alter brain neurochemistry and subsequent behavior relevant to the dopaminergic system. Indeed, the vagus nerve serves as a primary mediator in the gut–brain crosstalk to influence central and peripheral dopamine concentrations. For example, vagal nerve stimulation has been shown to induce dopaminergic activation and influence dopamine concentrations in the CNS [51]. The importance of vagal communication between the ENS and CNS has been well documented mainly through rodent models that undergo vagotomy. Vagotomized mice exhibit hampered neurogenesis regulation and decreased levels of brain-derived neurotrophic factor (BDNF) mRNA expression throughout the hippocampus [52]. BDNF has been shown to exert neuroprotective effects against dopamine neurons degeneration [53]. In addition to vagal-mediated pathways, the bidirectional communication between gut microbes and the CNS is facilitated through SCFAs, the immune system, and the hypothalamic–pituitary–adrenal (HPA) axis. Together, these modes of interaction of the microbiota-gut-brain axis link cognitive, emotional, and reward centers in the brain with visceral signals from the gut.

### 2.1. Short-Chain Fatty Acids and Dopamine

Gut microbiota metabolize complex, indigestible carbohydrates in the large intestine through anaerobic fermentation of dietary fibers and resistant starch to yield short-chain fatty acids SCFA The three well-known, main SCFAs produced in the colon as the end products of soluble fiber fermentation are butyrate (C4), propionate (C3), and acetate (C2) [54]. Their concentrations vary among individuals as a function of the abundance of specific microbes and have been shown to control and protect against systemic inflammation and maintain intestinal barrier integrity through gut–brain communication [55,56,57]. For example, inoculation of germ-free mice (GF) with *Clostridium butyricum* and *Bacteroides thetaiotaomicron* reduced blood–brain barrier (BBB) permeability by upregulating expression of brain tight junction proteins, occluding zonulin and claudin-5 through the neuroactive potential of SCFAs [58]. Likewise, significantly elevated claudin-5 serum levels have been found in individuals diagnosed with obsessive compulsive disorder, while decreased tight junction protein expression can lead to neuroinflammation [59], suggesting an important role of SCFAs in optimizing tight junction protein levels in endothelial tissues of the brain.

#### Butyrate and Dopamine

Butyrate has significant associations with gastrointestinal health, neurotransmitter concentrations, and gut–brain communication. Colonic enterocytes utilize butyrate produced by intestinal microbes from dietary fiber as their major energy source [57]. Firmicutes are major producers of butyrate with *Clostridium, Rumminococcus, Eubacterium,* and *Faecalbacterium* being notable genera largely contributing to its synthesis [60]. Butyrate’s intrinsic histone deacetylase (HDAC) inhibitor activity affects neurotransmitter levels and is therefore a key component of gut–brain signaling. For example, the HDAC inhibitory behavior of sodium butyrate has been shown to exert beneficial effects in neurotoxicity-induced rats by improving locomotor symptoms and increasing striatal dopamine [61]. It was shown that H3 histone acetylation was simultaneously increased when beneficial effects on locomotor symptoms and striatal dopamine levels became noticeable. HDACs remove acetyl groups from histones, further condensing DNA into a heterochromatin form. This makes DNA less accessible to transcription factors and other gene regulatory proteins. In addition, butyrate is a potent inhibitor of heterochromatin formation and functions to relax chromatin, leading to significant roles in epigenetic modulation. The HDAC inhibitory activity of sodium butyrate has been shown to reduce oxidative stress, exert beneficial effects on α-synuclein damage, and rescue dopaminergic cells in Parkinson’s disease (PD) models [62]. Recently, HDAC inhibitors have also been used as neuropharmacological agents to attenuate symptoms of glioblastomas [63]. Furthermore, sodium butyrate infusion attenuated neuronal apoptosis in mice with middle cerebral artery occlusions [64]. Together, these studies indicate possible therapeutic implications of butyrate’s intrinsic HDAC inhibitory activity on neurological disorders and neurotransmitter concentrations.

SCFAs can also incorporate colonic dopamine and produce serotonin through G-protein-coupled receptor (GPCR)-mediated pathways [42,65]. Two GPCRs that are expressed on enteroendocrine cells, including enteric and sympathetic neurons, are GPR41, also known as FA3R, and GPR43, also known as FA2R [66]. FA3R and FA2R are preferentially activated by butyrate and acetate, respectively. Study findings show protective effects of butyrate through FA3R signaling against salsolinol (SALS)-induced toxicity, a selective neurotoxin affecting dopaminergic neurons. When challenged with a selective FA3R antagonist, the protective effects of butyrate were significantly reduced [67], supporting the importance of FA3R/butyrate signaling in attenuating the progression of neurodegenerative disorders. Antagonism of FA3R diminished the positive effects of restoring the viability of neuroblastoma-derived dopaminergic cells via sodium butyrate administration in an ethanol-induced toxicity model [68]. Overall, these findings show that butyrogenic microbes play a major role in maintaining adequate dopamine concentrations by protecting against dopaminergic neuronal loss.

Additionally, sodium butyrate reduced degeneration of dopaminergic neurons, improved locomotor impairment, and elevated dopamine levels in a Drosophila transgenic model of PD [69]. A recent study supports these findings and shows that sodium butyrate administration in rodent models of PD reduces motor deficits while elevating dopamine levels and increasing histone acetylation [70]. Although much of the literature supports the beneficial effects of butyrate on the dopaminergic pathways, conflicting studies still exist. Qiao et al. concluded that sodium butyrate exacerbated motor symptoms, accelerated the loss of dopamine levels, and reduced tyrosine-hydroxylase-positive dopaminergic cells [71]. This study evaluated the effects of sodium butyrate after 1 week of treatment in neurotoxicity-induced mice, while other studies showing beneficial and contradictory findings examined the effects after 2 to 3 weeks of treatment [61,70,72]. This suggests that butyrate either can take time to exert its beneficial effects or that dopamine levels can be recovered after neurotoxicity induction.

Furthermore, butyrate has been shown to stimulate the production of gut hormones via enteroendocrine cells such as glucagon-like peptide 1 (GLP-1) and peptide YY (PYY), while also increasing GLP-1 receptor (GLP-1R) expression in the brain [72]. The increase in GLP-1R expression mitigated behavioral impairments in rodents. GLP-1 analogs and GLP-1R agonists have been shown to enhance synaptic membrane expression of DAT in the forebrain lateral septum [73]. More recently, GLP-1 analogs that cross the BBB and bind brain GLP-1R such as liratuglide have been found to improve cognition and enhance long-term potentiation [74]. Liratuglide-treated groups also showed significantly increased concentrations of neurotransmitters including dopamine in hippocampal tissue. Similar beneficial results were obtained with exenatide, a GLP-1R agonist, showing enhanced dopamine midbrain function and behavioral improvement [75]. Several butyrogenic microbial species can naturally enhance GLP-1 levels and brain GLP-1R expression to positively affect dopamine concentration and improve neurobehavioral deficits. Recent work on the potential use of synthetic GLP-1 agonists as a treatment for drug addiction [76,77] further highlights the importance of microbiota in the regulation of dopamine functions in health and disease via the GLP-1 system.

When oral butyrate tablets were administered, there were notable changes in concentrations of key microbes such as *Bacteroides uniformis* and *Prevotella copri* that affected DAT binding [20]. Heart rate variance was also observed exclusively in the butyrate-treated group and not in controls, indicating that the vagus nerve is stimulated by neuroactive butyrate. Furthermore, several studies showed the importance of maintaining adequate butyrate concentrations in the gut as it relates to neurological disorders that affect behavior. In children with autism spectrum disorder, levels of butyric acid, along with acetic and propionic acid, were significantly decreased and dopamine metabolism disorder, characterized by decreased homovanillic acid (HVA) was present [78]. Probiotic introduction containing SFCA-producing gut microbes increased HVA and improved the dopamine metabolism deficit. Overall, gut microbiota that produce butyrate have a myriad of effects on dopamine and contribute significantly to attenuating deficits seen in neurodegenerative disorders.

### 2.2. Cytokines, Gut Microbiota, and Dopamine

Cytokines appear to play an important role in the proposed regulatory axis between gut microbes, the nervous system, and the immune system. Regulation of dopamine has been shown to be associated with cytokine release and vice versa. For example, dopamine released in the gut due to microbial processing can transcriptionally regulate the production of certain cytokines such as ILF-4 and IFN-gamma [37]. In dysbiosis and dopamine deficiency, ILF-4 and IFN-gamma levels increase, potentially activating immune response through natural killer cells (NKT) [37]. Mice treated with an anti-bacterial cocktail showed markedly reduced levels of tyrosine hydroxylase both at the mRNA and protein levels within the small intestines. Subsequent restoration of gut microbiota following the anti-bacterial cocktail resulted in the recovery of intestinal tyrosine hydroxylase. The mechanism by which peripheral dopamine acts to decrease cytokine levels has been attributed to the activation of D1-like receptors on hepatic invariant NKT cells that play a role in regulating liver immunity. Additionally, the introduction of IFN-α for 4–6 weeks changes presynaptic dopamine function and decreases dopamine synthesis/release in the basal ganglia of patients with chronic hepatitis C virus [79]. These findings were associated with behavioral changes including depression and fatigue, indicating the widespread consequences of prolonged cytokine exposure in the body. It may also suggest that gut microbes promoting cytokine-mediated inflammation can alter striatal dopamine function by reducing dopamine production or release.

### 2.3. Hypothalamic–Pituitary–Adrenal Axis and Gut Microbiota

The hypothalamic–pituitary–adrenal (HPA) axis plays a dynamic role in the stress response and gut–brain communication [80]. Studies from GF rodents indicate that lack of gut microbiota is associated with a hyperreactive HPA axis in response to stress. This effect can be partially normalized by colonizing fecal matter with the normal flora of healthy rodents, indicating the importance of microbiota in maintaining homeostasis in the HPA axis [81]. Similarly, intestinal microbes have been shown to normalize the stress response by affecting HPA axis gene expression in conditions of chronic stress. For example, gut microbiota downregulate the FK506 binding protein 5 (Fkbp5) gene, which encodes a protein that regulates the negative feedback loop by reducing cortisol affinity for glucocorticoid receptors [82]. Thus, in the absence of gut microbiota, negative feedback becomes dysregulated, and an exaggerated HPA response ensues. Although gut microbiota function to regulate stress response, stress-inducing events confer detrimental effects to the normal intestinal flora. As such, hyperactivation of the HPA axis leads to gut microbiota dysbiosis through production of several proinflammatory cytokines such as IL-1β, IL-6, and TNF [80]. This can result in neurotransmitter dysregulations, leading to unfavorable behaviors via gut brain crosstalk, some of which can be restored through the administration of beneficial gut microbes. For example, administration of *Lactobacillus plantarum PS128* to GF mice significantly increased dopamine levels and improved anxiety-like behaviors [83], demonstrating how bidirectional gut–brain communication through the HPA-axis affects neurotransmitter release and activity.

## 3. Gut Microbes Effects on Dopamine

Gut microbiota are major contributors to dopamine bioavailability in the enteric and central nervous systems. The following subsections depict how key gut microbes such as *Prevotella, Bacteroides, Lactobacillus, Bifidobacterium, Clostridium, Enterococcus,* and *Ruminococcus* impact dopaminergic pathways.

### 3.1. Prevotella, Bacteroides, and Dopaminergic System

*Prevotella* and *Bacteroides* are microbial genera belonging to the phylum Bacteroidetes that have been shown to improve glucose metabolism, degrade a variety of plant polysaccharides, and produce favorable neuroactive SCFAs and vitamins that are necessary to promote gut health [84,85]. In addition to these symbiotic effects, metabolites produced by these microbes have been associated with dopamine functioning through modulation of dopaminergic synaptic cleft activity. In a recent study, Hartstra et al. showed that administration of *Bacteroides uniformis* through fecal microbiota transplant increased striatal DAT binding, while *Prevotella copri* was inversely correlated with transporter binding [20]. DAT, a presynaptic membrane protein present in dopaminergic terminals to regulate synaptic and extracellular dopamine is a critical modulator of dopaminergic tone within the CNS. Striatal dopamine binding to DAT within the synaptic cleft of dopaminergic neurons allows for recycling and storage of dopamine within vesicles in the presynaptic terminals for subsequent release. Although the mechanisms by which *Bacteroides uniformis* and *Prevotella copri* alter striatal binding are not clear, neuroactive metabolites are obvious candidates for modulating striatal DAT expression via activation of vagal afferents projecting to the nigrostriatal pathway within the synaptic cleft of dopaminergic neurons (Figure 1).

Furthermore, it was found that an abundance of *Prevotella* led to increased plasma concentrations of ghrelin, a gut-peptide orexigenic hormone that regulates satiety by acting on the VTA [86]. The VTA is a convergence point between ghrelin and dopamine, both of which are important in mediating reward pathways. As such, ghrelin was shown to recruit dopaminergic neurons by either direct activation or through indirect modulation of GABA neurons [87]. Systemically injected ghrelin has also been shown to stimulate motor activity and increase dopamine levels in rodent models. This suggests that *Prevotella* may indirectly recruit dopaminergic neurons by increasing ghrelin concentrations. Together, these findings indicate that the presence of *Prevotella* and *Bacteroides* in the intestinal tract does affect dopaminergic bioavailability either through direct actions on DAT binding or indirectly through alteration of gut-peptide hormone levels.

**Figure 1 biomedicines-10-00436-f001:**
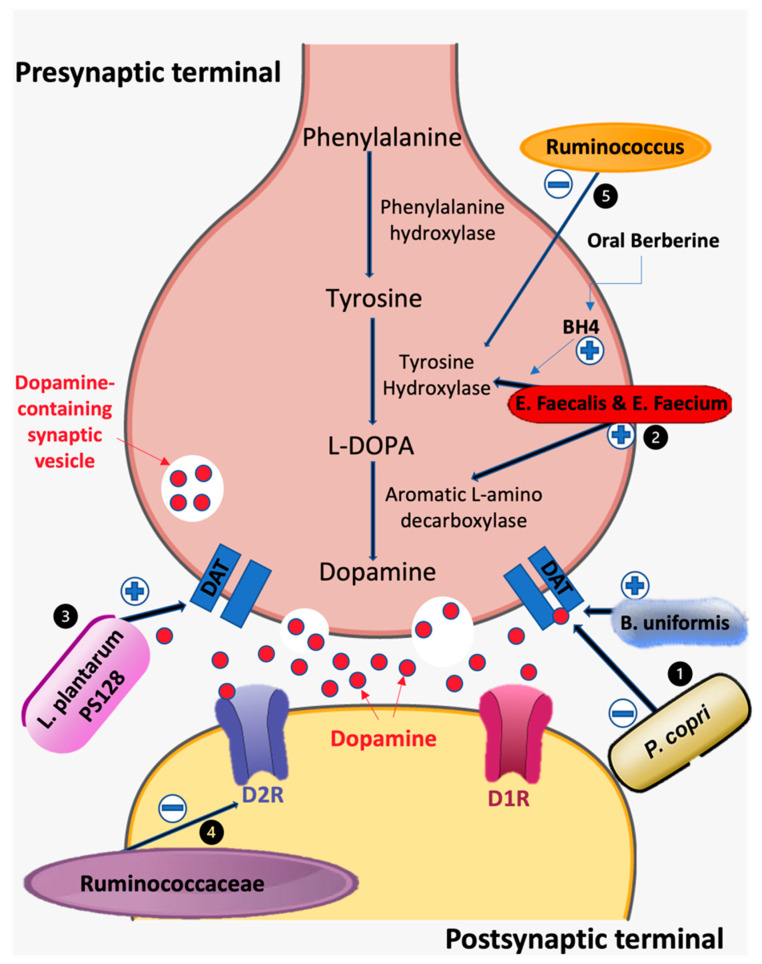
Effects of gut microbiota on the dopaminergic synaptic cleft and dopamine metabolism. (1) *Bacteroides uniformis* upregulates DAT/Dopamine binding efficiency. *Prevotella copri* downregulates DAT binding efficiency [20]. (2) *Enterococcus faecalis* and *Enterococcus faecium* demonstrate tyrosine hydroxylase and aromatic L-amino acid decarboxylase activity. Oral berberine synthesizes cofactor BH4, allowing enhanced tyrosine to dopamine conversion [35]. (3) *Lactobacillus plantarum PS128* increases DAT expression [88]. (4) *Ruminococcacae* downregulates D2R expression [32]. (5) *Ruminococcus* is correlated with decreased tyrosine hydroxylase activity [89]. Abbreviations: DAT, dopamine transporter; BH4, tetrahydrobiopterin; D1R, dopamine 1 receptor; D2R, dopamine 2 receptor.

### 3.2. Lactobacillus, Bifidobacterium, and Dopaminergic System

*Lactobacillus* and *Bifidobacterium* are two microbial genera commonly used as probiotics that belong to the phyla Firmicutes and Actinobacteria, respectively. Administration of *Lactobacillus* has been shown to reduce depression- and anxiety-like behaviors in mice whereas oral administration of *Bifidobacterium* reduced exaggerated HPA stress response [90]. Likewise, patients with psychiatric conditions display a lower abundance of *Bifidobacterium* and *Lactobacillus* when compared to healthy controls, suggesting that these probiotic genera modulate neurotransmission in the dopaminergic pathways [91]. For example, oral administration of *Lactobacillus plantarum PS128* for four weeks alleviated elevations in corticosterone, nigrostriatal dopaminergic neuronal death, and striatal dopamine reduction in 1-methyl-4-phenyl-1,2,3,6-tetrathydropyridine (MPTP)-induced PD in mouse models. MPTP is a neurotoxin that stimulates dopaminergic cell death in substantia nigra resulting in permanent manifestations of clinical symptoms in PD. Furthermore, PS128 administration was found to diminish MPTP-induced oxidative stress and neuroinflammation in the nigrostriatal pathway, supporting its function in dopamine modulation [92]. Additionally, Liao et al. further demonstrated that the administration of psychobiotics containing PS128 can improve dopamine-related neurological disorders. In this study, PS128 was administered to ameliorate tic-like behaviors by using 5-HT2A and 5-HT2C receptor agonist 2,5-Dimethoxy-4-iodoamphetamine (DOI). Rats were administered PS128 orally for two weeks followed by two daily DOI injections. Analysis of the brain tissues showed that PS128 administration improved dopamine metabolism, increased DAT, and increased norepinephrine (Figure 1 and Figure 2) [88]. Taken together, these studies provide strong evidence for the positive effects of *Lactobacillus plantarum* PS128 on dopamine metabolism and function.

In a clinical randomized, double-blind, placebo-controlled trial, administration of *Lactobacillus plantarum DR7* for 12 weeks lowered stress and anxiety in stressed adults and also reduced plasma pro-inflammatory and increased plasma anti-inflammatory cytokines compared to the placebo group. Additionally, *Lactobacillus plantarum DR7* lowered plasma cortisol, dopamine β-hydroxylase and tyrosine hydroxylase, two important enzymes in dopamine metabolism. It has also been documented that dysregulation of dopamine β-hydroxylase can stimulate the stress response [93]. Therefore, administration of *Lactobacillus plantarum DR7* can help reduce the stress response by lowering dopamine β-hydroxylase and tyrosine hydroxylase levels (Figure 2). Similarly, the effects of *Bifidobacterium infantis* were studied in a rat maternal separation (MS) model by evaluating motivational state, cytokine concentrations, brain monoamine levels, and central and peripheral HPA responses in MS rats subjected to a forced swim test (FST). Control MS rats displayed immobility in the FST, decreased brain noradrenaline, elevated IL-6, and enhanced amygdala corticotropin-releasing factor mRNA levels. However, probiotic treatment normalized the immune response and noradrenaline concentrations and reversed behavioral deficits of MS rats [94]. This study further supports the idea that *Bifidobacterium infantis* has an impact on neural function and neurotransmitter activity.

**Figure 2 biomedicines-10-00436-f002:**
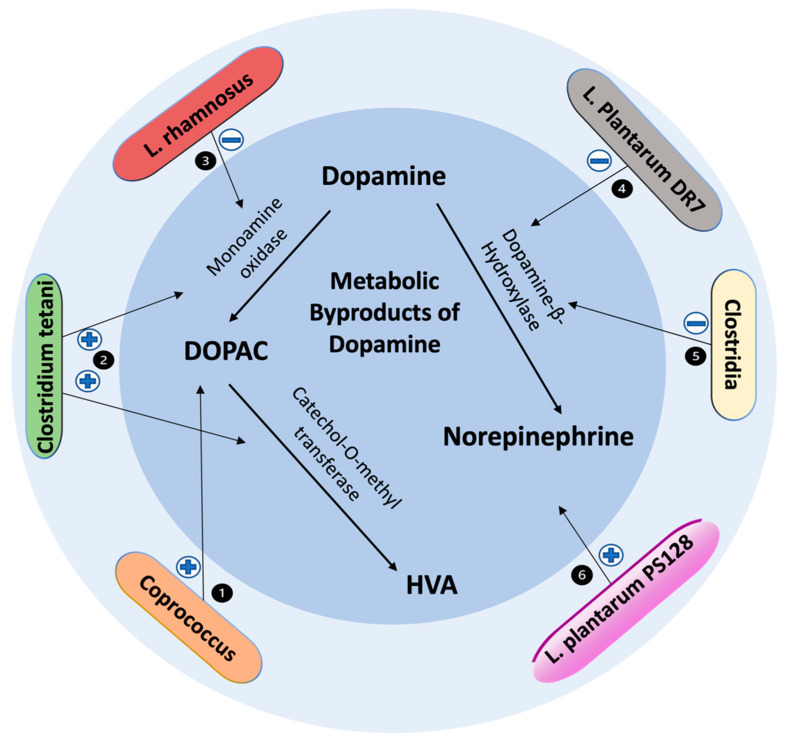
Effects of gut microbiota on metabolic byproducts of dopamine. (1) *Coprococcus comes* and *Coprococcus catus* are strongly associated with DOPAC synthesis potential [95]. (2) *Clostridium tetani* has degradative effects on dopamine, promoting degradation into HVA via the DOPAC intermediate [96]. (3) *Lactobacillus rhamnsosus* downregulates MAO [97]. (4) *Lactobacillus plantarum DR7* downregulates Dopamine β-hydroxylase [93]. (5) *Clostridia species* shown to downregulate Dopamine β-hydroxylase [98]. (6) *Lactobacillus plantarum PS128* administration improves dopamine metabolism and increases norepinephrine levels [92]. Abbreviations: MAO, monoamine oxidase; DOPAC, 3,4-Dihydroxyphenylacetic acid; HVA, homovanillic acid.

### 3.3. Clostridium and Dopaminergic System

*Clostridium* is a Gram-positive, anaerobic gut microbe that comprises roughly 180 species, making it one of the larger genera in microbial taxonomic classification to date. The *Clostridium* genus belongs to the phylum Firmicutes and contains species that have been shown to be the causative agents of pathogenic processes in humans [99]. Additionally, certain species in the *Clostridium* genus, such as *Clostridium tetani,* have been described to have degradative effects on dopamine. For example, when biogenic amines were placed in a medium containing *Clostridium tetani,* dopamine was metabolized into significant amounts of HVA via the DOPAC intermediate (Figure 2) [96]. HVA is an end product of enzymatic degradation of dopamine and coincides with decreased dopamine levels. Additionally, metabolites produced by *Clostridia* inhibit dopamine β-hydroxylase, thus blocking the conversion of dopamine to norepinephrine [98]. This causes a significant increase in dopamine levels while indirectly decreasing norepinephrine. The excess dopamine metabolites lead to toxic accumulation in the cytoplasm causing oxidative damage due to the depletion of glutathione in the brain. In turn, the oxidative stress produces a variety of detrimental effects, including apoptosis of dopaminergic neurons and a build-up of products such as α-synuclein protofibrils that are contributory to the pathogenesis of neurodegenerative disorders. These findings demonstrate how neuropathogenic species within the *Clostridium* genus can exert unfavorable effects on the enzymes involved in dopamine metabolism.

Notwithstanding the above findings, several studies have been shown to have beneficial probiotic-like effects of *Clostridium* species on dopamine concentration through their endospore-forming capacity [100]. *Clostridium butyricum* forms spores that germinate in the intestinal tract and lead to the production of considerable amounts of SCFAs, particularly butyrate [101]. These beneficial outcomes were demonstrated in a recent study, where administration of *Clostridium butyricum* to piglets exhibiting weaning stress was significantly associated with increased hypothalamic concentrations of dopamine [36]. This was attributed to high butyrate concentrations observed in weaning piglets supplemented with *Clostridium butyricum*. As noted above, butyrate can cross the BBB to influence activity in the brain and affect microglial function and development. Thus, sodium butyrate decreases neuroinflammation and enhances neuroprotective function on cortical neurons [102], including direct effects on hypothalamic dopamine levels.

In addition to *Clostridium butyricum*, other species such as *Clostridium coccoides* and *Clostridium leptum* have been shown to have β-glucuronidase enzymatic activity that assists in the production of free gut catecholamines [18]. When comparing GF mice with select-pathogen-free (SPF) mice, SPF mice were shown to have considerably higher amounts of dopamine, epinephrine, and norepinephrine levels compared to GF mice. Peripheral catecholamines are commonly found to be conjugated with either glucuronide or sulfate and when conjugated are considered biologically inactive. Most dopamine was identified in the “free” form in the cecum and colon along with substantial amounts found in the ileum. In SPF mice, gut catecholamines, including dopamine, were found in the biologically active free form, while in GF mice, the gut catecholamines were in the biologically inactive, conjugated form. This indicates that *Clostridium coccoides* and *Clostridium leptum* contained β-glucuronidase enzymatic activity to convert peripheral dopamine into its active form. Thus, *Clostridium* plays important roles in increasing the available, functional dopamine in the periphery as well.

More recently, a study by Chang et al. showed that decarboxylated neurotransmitter substrates, including dopamine, can be conjugated with fatty acids to produce fatty acid amides. *Clostridia* produce these fatty acid amides in the gut lumen either endogenously or by using exogenous substrates that are common in the human diet. They also act as ligands binding GPCRs, leading to downstream signaling effects that drive physiological processes [65]. Thus, *Clostridia*’s ability to conjugate dopamine-like substrates into fatty acid amides to stimulate GPCR signaling warrants further studies into its pathogenic and homeostatic effects. Nevertheless, it is apparent that the genus *Clostridium* contains species that exhibit differing effects on dopamine metabolism. These widespread findings merit further exploration into the potential role of *Clostridium* to modulate systemic effects through the incorporation of dopamine substrates and metabolites.

### 3.4. Enterococcus and Dopaminergic System

*Enterococcus* is another key genus of microbiota that is a part of the phylum Firmicutes, which contains species that have been shown to participate in dopamine production. Although the genus is part of the normal intestinal flora, it has been identified as an opportunistic pathogen. Its two major species, *Enterococcus faecalis* and *Enterococcus faecium***,** have been shown to cause urinary tract infections, endocarditis, and other hospital-acquired infections [103]. Notwithstanding, significant research has been conducted to further understand their functions and impact on dopaminergic pathways. For example, several studies have demonstrated that supplementation of L-dopa enables *Enterococcus faecium* to convert newly introduced L-dopa into dopamine in the GI tract [35]. Additionally, transplantation of both *Enterococcus faecalis* and *Enterococcus faecium* into a mouse model of PD was shown to drastically increase the amount of striatal dopamine, as confirmed by neuroimaging. More specifically, this study found that tyrosine hydroxylase activity was also carried by the two species of *Enterococci* and was further activated by oral administration of berberine. Berberine stimulates gut microbiota to synthesize the tetrahydrobiopterin (BH4) cofactor and enhance tyrosine conversion into L-dopa [104]. These recent advancements by Wang et al. identifying novel findings of tyrosine hydroxylase activity in *Enterococcus faecium* are a significant step in describing the dopamine synthesis capability of the *Enterococcus* genus (Figure 1). Together, these two studies highlight the enzymatic capacity of *Enterococcus faecium* and *Enterococcus faecalis* to convert tyrosine or L-dopa to dopamine.

Although *Enterococcus faecalis* has the enzymatic capabilities to produce dopamine, it appears that it can also deplete dopamine precursors in the GI tract. It is important to note that *Enterococci* preferentially convert tyrosine to tyramine via tyrosine decarboxylase enzymatic activity [105]. Indeed, a recent study showed that *Enterococcus faecalis* also efficiently decarboxylated L-dopa via a pyridoxal phosphate-dependent tyrosine decarboxylase, which was then further metabolized to tyramine [106]. The ability of *Enterococci* to metabolize the two essential dopamine precursors, tyrosine and L-dopa, into tyramine can become problematic for maintaining sufficient dopamine levels. Interestingly, Maini Rekdal et al. provided evidence that α-flouromethyltyrosine prevents the decarboxylation of L-dopa into tyramine by *Enterococcus faecalis.* By blocking the preferential conversion of L-dopa to tyramine via α-flouromethyltyrosine, it is plausible that the intrinsic enzymatic activity within *Enterococcus faecium* and *Enterococcus faecalis* can instead convert the dopamine precursors into dopamine. As such, *Enterococcus faecium* and *Enterococcus faecalis,* when used in conjunction with α-flouromethyltyrosine, can become candidates as probiotics due to their potential to synthesize dopamine and attenuate neurotransmitter imbalances seen in neurodegenerative disorders.

### 3.5. Ruminococcus and Dopaminergic System

*Ruminococcus*, an anaerobic, Gram-positive microbial genus belonging to the phylum Firmicutes has been associated with dopamine metabolism. Although *Ruminococcus* species produce SCFAs, which have neuroprotective effects on dopaminergic neurons, their role in mucin-degradation has opposite consequences. Degeneration of intestinal mucosa by *Ruminococcus* stimulates the production of cytokines, causing colon inflammation. A recent study showed the negative effects of *Ruminococcus* on dopaminergic neurons by assessing tyrosine hydroxylase activity in MPTP-neurotoxicity induced mice. MPTP neurotoxicity stimulates intestinal dysbiosis, increases *Ruminococcus* concentrations and causes degeneration of tyrosine hydroxylase-positive cells in the nigrostriatal pathway. Interestingly, consumption of Korean red ginseng by MPTP-treated mice caused attenuation of dopaminergic degeneration by decreasing the abundance of *Ruminococcus*. This was associated with increased tyrosine hydroxylase and thus increased dopamine levels [89]. Similarly, exercise-induced stress behavior also caused changes in the abundance of *Ruminococcus* gnavus, which also modulates intestinal mucus degradation and immune function [107]. Recent results from metagenomic analysis via shotgun sequencing of fecal samples from 49 children with a tic disorder (TD) and 50 matched healthy controls showed that gut microbiota from TD subjects had a higher abundance of *Ruminococcus lactaris* and *Ruminococcus gnavus* [108]. Clinical evidence also supports a dysregulation in dopamine levels in TD, though the exact effect on dopaminergic neurons, receptors, and transporters is less clear [109]. Still, it is evident that increased *Ruminococcus lactaris* and *Ruminococcus gnavus,* along with other gut microbial changes, confer neurotransmitter changes that are contributory to the negative effects seen in TD.

The expression of D1 and D2 receptors has been shown to be influenced by the composition of gut microbiota. For example, *Ruminococcaceae* and *Lachnospiraceae* have been associated with decreased D2 receptor mRNA expression in the dorsal striatum of alcohol-induced mice [32]. The D2 receptor exerts downstream G-α-inhibitory signaling effects, which inhibit adenylate cyclase and lower cAMP levels. Polymorphisms leading to decreased dopamine binding to D2 receptor haplotypes have been associated with dyskinesia [110,111], one of the classical symptoms of PD. Thus, it can be hypothesized that the abundance of *Ruminococcacae* seen in patients with PD causes downregulation of D2 receptor expression, manifesting in motor symptoms such as bradykinesia. This is an intriguing proposition that can be investigated in future studies.

## 4. Microbiota, Dopamine, and Parkinson’s Disease

Dopamine bioavailability in humans is significantly intertwined with psychiatric conditions [112]. Several studies associate changes in gut microbes and dopamine with the pathogenesis and clinical presentation of patients with PD that is caused by neurodegeneration of substantia nigra and midbrain nuclei dopaminergic neurons [113,114]. The pathophysiology of PD can be largely attributed to the accumulation of α-synucleins, which interfere with synaptic transmission in dopaminergic neurons resulting in reduced dopamine bioavailability [115]. Dopamine loss in the periphery manifests as GI malfunctions, including delayed gastric emptying and intestinal dysmotility. Additionally, the neurodegeneration of dopaminergic neurons in the nigrostriatal pathway leads to motor symptoms, including bradykinesia, resting tremors, and postural instability. These motor symptoms, termed parkinsonism, have been directly associated with changes in gut microbial composition. For example, elevated *Enterobacteriaceae* has been correlated with postural instability and gait abnormalities [85].

Some gut microbes have been shown to exert neuroprotective effects on dopaminergic neurons to attenuate dopamine loss. On the other hand, other microbes can exert negative effects by stimulating inflammatory responses through endotoxins to further deplete dopamine concentrations. Overall, beneficial microflora tend to be reduced in the parkinsonian gut, while levels of microbes that induce pathological processes are elevated. Trends that reflect the gut microbial composition of patients with PD have been identified in the literature. Although discrepancies do exist between studies, extensive microbial genotyping of the parkinsonian gut has been conducted and many analyses display comparable results. All six major bacteria phyla are affected in some way or another by PD and changes in gut microbial species may serve as prodromal biomarkers that may indicate the early onset of the disease [85]. Altered taxa of PD patients include, but is not limited to, an increase in *Akkermansia muciniphilia, Enterobacteriaceae, Eggerthella, Oscillibacter, Catabacter, Escherichia, Shigella, Megomanus, Lachnospiraceae,* and *Streptococcus* and decreases in *Roseburia, Coprococcus, Faecalibacterium*, and *Eubacterium biforme* [116,117,118,119].

Of the genera described extensively in this review, *Enterococcus, Bifidobacterium*, and *Rumminococcus* levels were found to be elevated while *Prevotella, Bacteroides,* and *Clostridium* concentrations were decreased in PD [120,121,122]. Findings regarding the effects of *Lactobacillus* appear to be less certain within the literature, with some reporting increases while others claim decreases in this microbial genus [121,123,124]. Together, these studies show that changes in these gut microbial genera contribute either positively or negatively to the pathophysiology of PD (Table 1). The possible mechanisms by which intestinal microbial dysbiosis and its resulting effect on dopamine bioavailability may lead to the pathogenesis of PD are highlighted in the following sections (Figure 3).

**Table 1 biomedicines-10-00436-t001:** Microbes’ effects in Parkinson’s Disease.

Phylum	Genus	Change	Effects on Parkinson’s Pathophysiology	References
Bacteroidetes	*Prevotella*	Reduced	Reduced secretion of neuroprotective hydrogen sulfide into the gut lumen and decreased intestinal motility	[117,118,121,125]
	*Bacteroides*	Reduced		[119,126]
Firmicutes	*Lactobacillus*	Reduced	Less neuroprotective effects, unable to rescue dopaminergic neuron loss, Inability to downregulate MAO-B, Reduced BDNF and tyrosine hydroxylase expression	[34,97,119,121]
	*Lactobacillus*	Elevated	Deconjugates neuroprotective bile acids, TDCA and UDCA	[118,124,127,128]
	*Clostridium*	Reduced	Increased α-synuclein accumulation	[117,124,128,129]
	*Enterococcus*	Elevated		[122]
	*Ruminococcus*	Elevated	*Ruminococcus Albus* reduces ROS species, increases BDNF effects, and has associations with decreased neuroprotective bile acids	[120,121,127,130]
Actinobacteria	*Bifidobacterium*	Elevated	Confers neuroprotective effects on dopaminergic neuron loss	[97,121]
Verrucomicrobia	*Akkermansia*	Elevated	Mucin-degrading genus, increases LPS and microglial activity	[116,117,118,131]

**Figure 3 biomedicines-10-00436-f003:**
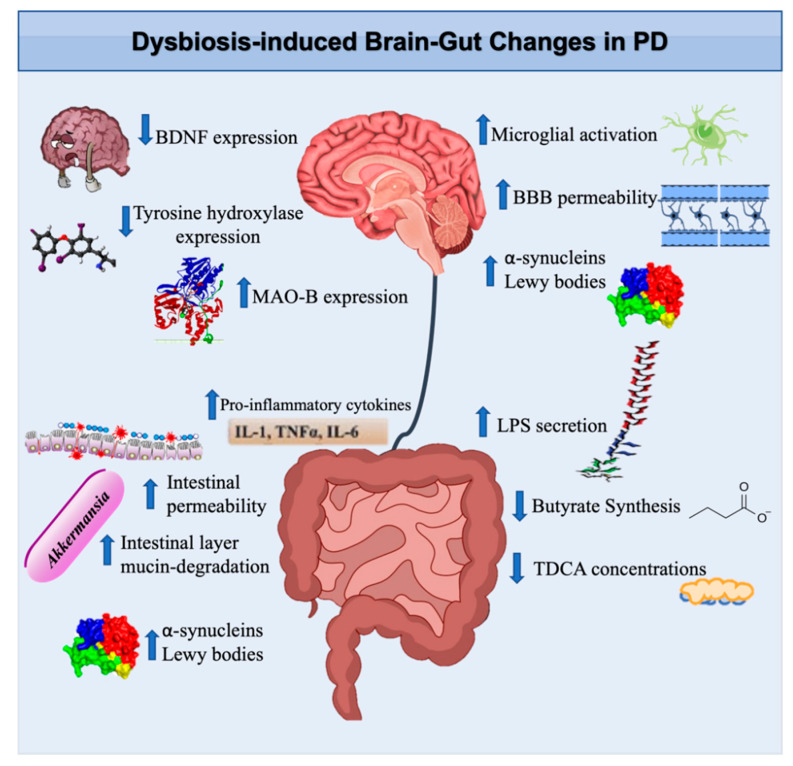
Gut microbial dysbiosis in Parkinson’s Disorder pathophysiology in the brain and GI tract. Patients with PD exhibit increased microglial activity [131], BBB permeability [58], α-synuclein aggregation [128], MAO-B expression [97], LPS secretion [131], intestinal permeability, intestinal layer mucin degradation, and pro-inflammatory cytokine release [131]. There is a decrease in BDNF expression and tyrosine hydroxylase expression [34], butyrate synthesis, and TDCA concentration [132]. Abbreviations: BDNF, brain-derived neurotrophic factor; BBB, blood–brain barrier; MAO-B, monoamine oxidase B; LPS, lipopolysaccharide; IL-1, interleukin-1; TNFα, tumor necrosis factor α; TDCA, taurochenodeoxycholic acid.

### 4.1. Gut Microbes and α-Synucleins in Parkinson’s Disease

Alpha-synucleins, a presynaptic protein, are key to the pathogenesis of PD, having been involved in the regulation of dopamine release and its synaptic vesicle pool. For example, α-synuclein knockout mice have fewer tyrosine hydroxylase cells, while endogenous mouse α-synuclein was associated with an increase in the number of tyrosine-hydroxylase-positive cells in the substantia nigra pedunculopontine nucleus [133], suggesting a close link between α-synucleins and dopamine levels. In PD, the α-synucleins become less soluble and aggregate with other proteins, inducing Lewy body accumulation within the substantia nigra [134]. The misfolded, insoluble α-synucleins confer toxic neurological consequences by interfering with the synaptic transmission of dopaminergic neurons. Thus, the aggregation of α-synucleins is a pathological hallmark in PD.

Interestingly, the process of Lewy Body formation through α-synuclein aggregation may begin in the GI tract and ascend via vagal afferents into the dorsal motor nucleus of the vagus and SNpc [135,136]. Braak’s hypothesis postulates that the initiation of PD occurs in the gut, presumably through the buildup of α-synucleins within enteric neurons [137]. Accumulation of α-synucleins in the ENS, specifically within the colonic submucosa, was shown through immunohistochemistry studies in Parkinson’s patients [138]. A study by Yang et al. further supports these findings by stating that rotenone neurotoxicity-induced mice exhibited higher colonic α-synuclein levels prior to the increase in its amounts in the midbrain. Additionally, microbial dysbiosis, produced through the use of rotenone, preceded the classic neurological symptoms of PD [124]. Rotenone is an electron transport chain inhibitor that mimics the parkinsonian-gut promoting dopaminergic neuron loss and formation of α-synucleins. In the same study, the rotenone-treated mice displayed increased *Lactobacillus* and decreased *Clostridium*. The overall gut microbe composition in the neurotoxicity-induced mice shows a higher Firmicutes/Bacteroides ratio, which is consistent with many pathological conditions.

O’Donovan et al. showed that overexpression of α-synucleins in the SNpc results in lowered submucosal tyrosine hydroxylase intensity, decreased submucosal neuronal density, and changes in microbial composition [127]. This increase in α-synucleins was associated with an increase in fecal bile acids. Ursodeoxycholic acid (UDCA) and taurochenodeoxycholic acid (TDCA) are two conjugated bile acids that exhibit neuroprotective effects by preventing dopaminergic cell death in rodent models [139]. Two notable changes in the gut microbial composition seen in O’Donovan’s study were *Rumminococcus* and *Lactobacillus* genera. The association between *Rumminococcus* and the bile acids was less clear. However, a significant correlation was found with *Lactobacillus* due to its ability to deconjugate bile acids into their unconjugated form via bile salt hydrolase activity [140]. Bile acids are usually recycled continuously and reabsorbed in the terminal ileum [141]. Fluctuations in *Lactobacillus* concentrations as seen in PD can lead to deconjugation of important bile acids such as TCDA reducing its concentration in the GI tract, leading to reduced neuroprotection and dopaminergic cell death.

### 4.2. Gut Microbes and Neuroprotective Effects

Several intestinal microbial species have prominent roles in neuroprotection of dopaminergic neurons. Introduction of a probiotic containing *Lactobacillus rhamnosus, Lactobacillus acidophilus,* and *Bifidobacterium animalis* rescued dopaminergic neuronal loss in the nigrostriatal pathway of neurotoxicity-induced rats. Of the three species, *Lactobacillus rhamnosus* is the major microbial species that upregulated key neurotrophic signals and downregulated monoamine oxidase (MAO-B), an enzyme known to break down dopamine [97]. The elevated BDNF and glial-cell-line-derived neurotrophic factor (GDNF) were shown to regulate the differentiation of neurotransmitters and protect against the degradation of dopaminergic neurons [53]. Reduced BDNF mRNA expression in the substantia nigra has also been shown in PD [142]. Probiotic administration resulted in increased brain butyrate concentrations in neurotoxicity-induced rats providing strong evidence that butyrate influences attenuation of the dopaminergic neuron loss.

Liu et al. offered another intriguing explanation behind butyrate’s neuroprotective mechanism against PD, suggesting that sodium butyrate upregulates tight junction proteins to attenuate BBB disruption in neurotoxicity-induced mice. In the same study, increased expression of colonic and brain GLP-1 receptors was also present [72]. GLP-1 analogs and GLP-1R agonists have been shown to enhance synaptic membrane expression of DAT in the forebrain lateral septum [73]. This would, in turn, increase transport capacity, recycling, and vesicle storage of dopamine for later release. As a result, it is possible that the protective effects against dopaminergic degeneration seen in Liu et al.’s study can be attributed to increased GLP-1 and GLP-1R expression. Likewise, a recent study showed that gut-microbiota-derived propionic acid could mediate the neuroprotective effect of osteocalcin in a 6-hydroxydopamine-induced mouse model of PD. Considering the protective role of propionic acid, it is likely that the reported low levels of propionic acid in the feces of PD patients may play a role in the pathogenesis of PD [143].

*Lactobacillus plantarum* also was shown to exert similar neuroprotective effects in conditions of oxidative stress. Cheon et al. created an environment of oxidative stress known to exacerbate neurodegenerative diseases before injecting the heat-killed *Lactobacillus plantarum* strain into a conditioned media. There was an increased mRNA expression of BDNF and tyrosine hydroxylase [34], indicating that the microbe stimulates attenuation of dopaminergic neuronal loss. This experiment followed a similar model of an earlier study that investigated the neuroprotective effects of *Lactobacillus buchneri* [144] and *Rumminococcus albus* in conditions of oxidative stress [130]. The results were comparable, showing that heat-killed *Lactobacillus buchneri* and *Rumminococcus albus* increased BDNF expression and reduced reactive oxygen species (ROS) to exert neuroprotective effects on neurons. However, tyrosine hydroxylase expression was not elevated in *Rumminococcus albus* and *Lactobacillus buchneri.* It is also important to note that *Lactobacillus plantarum, Rumminococcus albus,* and *Lactobacillus buchneri* diminished the Bax/Bcl-2 ratio induced by the oxidative environment. Increased Bax/Bcl-2 stimulates cellular apoptosis [145]. As such, it appears that the protective mechanisms of these microbes in PD are present at the cellular and neuronal levels.

### 4.3. Lipopolysaccharides

Liposaccharides (LPSs) are endotoxins naturally found in the outer membrane of cell walls in Gram-negative bacteria. LPSs promote inflammation by binding Toll-like receptor 4 (TLR-4), leading to transcriptional upregulation of pro-inflammatory cytokines. In the short-term, systemic inflammation through LPS-mediated pathways has been found to elevate dopamine concentrations in the dorsal striatum [146]. These findings are consistent with studies showing that IFN-α administration in a primate model for two weeks enhances dopamine release. However, chronic exposure causes an opposite effect, reducing striatal dopamine release as well as D2 receptor binding [147]. Furthermore, LPS administration can influence brain activity and is correlated with the development of neurodegenerative pathologies. LPS produces ROS through NADPH oxidase activity, resulting in microglial activation and dopaminergic neuronal loss [148]. It is also documented that tyrosine hydroxylase activity is reduced in the substantia nigra, leading to noticeable motor dysfunction. This is in line with studies suggesting that LPS administration increases intestinal permeability, allowing endotoxins to travel towards the midbrain, causing neuroinflammatory effects [131], increased mid brain TLR-4 reactivity, activation of microglia, and dopamine loss in rotenone-induced mice. These findings were associated with increases in the abundance of *Akkermansia,* a Gram-negative, mucin-degrading microbe shown to be elevated in the parkinsonian gut [116], leading to endotoxemia-induced chronic inflammation.

As mentioned, α-synucleins aggregation has been associated with bacterial endotoxin [149]. The mucin-degrading capacity of *Akkermansia* beneficially assists in maintaining intestinal barrier integrity at moderate to low levels. When elevated, reduced tight junction protein expression leads to intestinal leakiness. LPS can escape through the more permeable intestines into enteric neurons and systemic circulation [150]. As a result, LPS can then exert widespread effects that promote α-synuclein aggregation and dopaminergic neuronal loss. Additionally, LPS induces amyloidogenesis through the stabilization of α-helical intermediates. LPS and α-synucleins examined via microscopic imaging were shown to undergo a series of rapid nucleation events, leading to the formation of α-synuclein/LPS fibrils [151] that can worsen PD pathogenesis. Taken together, these findings suggest that increased LPS activity exerts detrimental effects on dopamine bioavailability by producing ROS, increasing intestinal permeability, activating microglia, and promoting the formation of α-synucleins leading to dopaminergic neuronal loss.

### 4.4. Gut Microbes, Dopamine, and Intestinal Dysmotility

The literature suggests that clinical GI manifestations often precede motor complaints in PD. For example, the prodromal phase of PD includes symptoms of delayed gastric emptying and constipation that are correlated with dopaminergic neuronal loss and α-synucleins accumulation in the periphery [152]. These findings can be an important sign in diagnosing the early onset of PD. In general, higher Firmicutes/Bacteroides ratios have been correlated with intestinal dysmotility. More specifically, the Firmicutes subgenera, *Clostridium,* and subfamily, *Rumminococcacae,* were shown to be elevated in patients with functional constipation, while *Prevotella* levels were decreased [153]. *Prevotella* has been shown to secrete hydrogen sulfide into the gut lumen. In rodents, hydrogen sulfide exerts protective properties against degeneration of tyrosine hydroxylase-containing dopaminergic neurons [154], which is correlated with manifestations of intestinal dysmotility in PD [155]. Thus, some studies have proposed that decreased concentrations of the hydrogen-sulfide secreting *Prevotella* contribute to GI symptoms such as constipation secondary to lower peripheral dopamine levels [125]. Although dopamine is known to inhibit upper GI motility and gastric emptying, it also plays a stimulatory role in lower GI motility, primarily in the colon, ameliorating the discomfort from constipation [156]. Therefore, lesser concentrations of peripheral dopamine would reduce lower intestinal motility, causing constipation. As such, reduced *Prevotella* concentrations present in the parkinsonian gut and associated reduction of hydrogen-sulfide secretion would be a factor in motility-related symptoms seen in PD. These conclusions are further supported by an analysis that identifies *Prevotella* as the only taxon that was significantly reduced in PD patients and was correlated with ROME III constipation scores [126]. Furthermore, recent findings indicate that *Prevotella*-enriched enterotypes, as opposed to Firmicute-enriched enterotypes, lead to much lower constipation severity in PD [157]. This concurs with data suggesting that *Prevotella* remains an important microbial genus for the prevention of common GI symptoms in neurodegenerative disorders.

Although most studies support the idea that reduced *Prevotella* plays an important role in the parkinsonian gut, conflicting studies exist. An analysis of a transgenic rodent model with similar pathophysiological components to PD showed a transient increase in *Prevotella* at 20–24 weeks as compared to age-matched controls. At similar time points, there was a decrease in dopamine production both in the midbrain and in the gut [158]. This was correlated to an early onset of GI symptoms associated with PD. Moreover, another study claimed that Unified Parkinson’s Disease Rating Scores (UPDRS) were not associated with *Prevotella*, in contrast with previous work [159]. Notwithstanding, the preponderance of studies show that *Prevotella* richness is commonly reduced in the parkinsonian gut.

Furthermore, species within the *Lactobacillus* and *Bifidobacterium* genera that are commonly used in probiotics to treat intestinal dysmotility were unchanged in constipated patients [153]. However, introducing *Lactobacillus* and *Bifidobacterium* as a probiotic improved stool frequency and reduced the average gut transit time from 135 h to 77 h in PD patients [160]. These effects have been attributed to SCFAs that increase GI motility through the regulation of enteric neurons. Since SCFAs increase α-synuclein deposition in enteric neurons [122], increased abundance of SCFA-producing microbial species attenuates constipation symptoms over time by reducing α-synuclein aggregation. Additionally, *Lactobacillus* deconjugates bile acids such as TDCA in the GI tract, which has been shown to inhibit intestinal motility [132] in addition to its neuroprotective effects. Thus, it is possible that the introduction of *Lactobacillus* can restore GI motility by deconjugating TDCA and removing its inhibitory effect. Overall, intestinal dysbiosis resulting in decreased *Prevotella, Lactobacillus,* and *Bifidobacterium* concentrations is a major factor in the intestinal dysmotility seen in patients with PD.

## 5. Conclusions and Perspectives

Substantial evidence supports the involvement of microbiota-gut-brain signaling in dopamine release, synthesis, and bioavailability. In this review, we focused on key microbial genera, *Prevotella, Bacteroides, Lactobacillus, Bifidobacterium, Clostridium, Enterococcus*, and *Ruminococcus,* that are intricately intertwined with dopaminergic pathways via myriad effects on dopamine precursors, enzymes, receptors, transporters, and metabolites. States of intestinal dysbiosis involving these key genera disrupt microbiota-gut-brain signaling, leading to dopaminergic deficits that manifest in neuropathological conditions like PD. Overall, the literature continues to support the notion that pathophysiological effects in PD begin within the GI system. Notable pathological markers in the GI tract that are attributed to gut microbial changes include increased intestinal inflammation, mucin-layer degradation, LPS secretion, and α-synuclein accumulation and decreased butyrate synthesis and neuroprotective bile acid production. Vagus-mediated pathways communicate these pathological manifestations to the CNS, leading to neurodegenerative effects on dopaminergic neurons. Consequently, expression of BDNF, MAO-B, and tyrosine hydroxylase is negatively affected by decreased BBB integrity and increased neuroinflammation through microglial activation and ROS production. Although significant advances have been made to elucidate the associations between gut microbiota, dopamine, and related pathophysiology, there remains much to be learned. Many studies have been conducted in animal models that have been developed to mimic a parkinsonian patient through MPTP or rotenone-induced neurotoxicity. PD etiology is a progressive neurodegenerative disease that involves contributory lifestyle, genetic, and socioeconomic factors which are difficult to replicate in animal studies. Food consumption and environmental variability are associated with differing gut microbiomes worldwide [161]. For example, consumption of the Western diet has been associated with specific microbial enterotypes that can predispose individuals to various pathologies. Elevated *Ruminococcus* resulting from a Western diet results in pro-inflammatory behavior underlying the pathophysiology seen in neurodegenerative disorders such as PD [162]. Metagenomic sequencing of gut microbiota following Western diets also reveals a limited prevalence of *Prevotella* that is associated with decreased DAT binding affinity. Further, the results from twin studies demonstrate how gut microbiome composition is largely connected to non-genetic factors [163,164]. Thus, it is plausible that individuals from different communities are more susceptible to developing dopamine-related pathologies based on their gut microbial makeup shaped by diet and environment. Controlling for these variables in future studies is critically important and will help in our understanding of gut microbiota involvement in neurodegenerative etiology. Furthermore, some animal studies did not consider the different time points and progression of the parkinsonian state when measuring dopaminergic changes. Early PD and late PD have different clinical manifestations and may reflect the various effects on the dopaminergic pathways and other pathological changes detailed in this review. Nevertheless, the progress thus far establishing the link between gut microbiota and neurodegenerative disorders is a step forward in our understanding of the complex mechanisms underlying these pathologies.

## Data Availability

Not applicable.

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
