# Peer review of "Role of Microbiota-Gut-Brain Axis in Regulating Dopaminergic Signaling"

_biomedicines, 2022, doi:10.3390/biomedicines10020436_

Round 1
Reviewer 1 Report
In this review, Sevag and colleagues discuss the role of microbiota-gut-brain axis in regulating dopaminergic signaling. The manuscript is well written. I have only some suggestions that may improve this review:
- The manuscript is very long and in my personal opinion in several parts the Authors lose focus. As the Authors are discussing the link between the gut brain axis and the dopaminergic signaling, other statements that do not belong to this link should be deleted in order to shorten the manuscript and make it more easy to read.
- The Authors should add more general information about the dopaminergic system. The description of the dopaminergic system is limited to few statements (lines 63-70). In this respect, the Authors might want to add (line 70, references 25-27) the following references PMID: 33167370; 31057408; 30648615), which link dopamine with reward, cognition and stress-related pathology.
- From a therapeutic perspective, the Authors should also discuss the fact that targeting the gut-brain axis to rescue dopaminergic dysfunctions might be an optimal therapeutic strategy also for other neuropsychiatric disorders, not only PD, which are characterized by dopaminergic dysfunctions (for example schizophrenia and PTSD, see please the references above).
Author Response
In this review, Sevag and colleagues discuss the role of microbiota-gut-brain axis in regulating dopaminergic signaling. The manuscript is well written. I have only some suggestions that may improve this review:
Thank you for your constructive and positive comments which helped improved our paper.
The manuscript is very long and in my personal opinion in several parts the Authors lose focus. As the Authors are discussing the link between the gut brain axis and the dopaminergic signaling, other statements that do not belong to this link should be deleted in order to shorten the manuscript and make it more easy to read.
Response: We have revised and shortened the manuscript significantly (by 8 pages) and removed extraneous information from the gut-brain axis and dopaminergic section. Also, at the suggestion of Reviewer #2 we’ve removed the section on Strategies to restore dopaminergic deficits (FMTs, WLS, probiotics). As a result, the number of references was reduced from 268 to 164.
The Authors should add more general information about the dopaminergic system. The description of the dopaminergic system is limited to few statements (lines 63-70). In this respect, the Authors might want to add (line 70, references 25-27) the following references PMID: 33167370; 31057408; 30648615), which link dopamine with reward, cognition and stress-related pathology.
Response: Thank you for providing these references. As requested, we have added the pertinent information to the corresponding sections related to reward, cognition, and stress-related pathologies and included the references.
From a therapeutic perspective, the Authors should also discuss the fact that targeting the gut-brain axis to rescue dopaminergic dysfunctions might be an optimal therapeutic strategy also for other neuropsychiatric disorders, not only PD, which are characterized by dopaminergic dysfunctions (for example schizophrenia and PTSD, see please the references above).
Response: As requested by Reviewer #2, the “restoration of dopaminergic deficits section” has been removed. The related sentences in the abstract, introduction and conclusion were also removed. We've adjusted figures and tables accordingly.
Reviewer 2 Report
Regarding the manuscript "The Role of microbiota-gut-brain axis in regulating dopamine signalling" the following issue should be addressed:
- The article is interesting, however there are multiple articles recently published on this topic.
- The length of the manuscript is quite long. All together, there are over 200 references.
- What is the purpose of putting together Parkinson's disease and Roux-en-Y gastric bypass?
- The role of microbiota is largely displayed and then treatment options (fecal transplantation, probiotics and weight loss surgery) are also discussed. Could these be two subjects for two different papers?
- What is the purpose of introducing chemical formulas in Figures?
- Figure 1 has a blurry aspect.
Author Response
The article is interesting, however there are multiple articles recently published on this topic.
Response: Thank you for your positive and constructive comments which helped improve our paper. In as much as there are few papers on the link between gut microbiota and dopaminergic system, most of them are either related to the pathophysiology of specific diseases such as depressive disorder, anxiety, Alzheimer’s disease; therapeutical approaches using gut microbiota in PD or more general papers on the role of gut microbiota in regulating neuronal circuits. Our paper covers a wider range of pertinent information to the readership by integrating recent advances on the role of microbiota-gut-brain axis, examining the relationship between key bacteria genera, their metabolites, their effects on dopamine, mechanisms of actions and implications in the pathophysiology of PD.
The length of the manuscript is quite long. Altogether, there are over 200 references.
Response. We have shortened the manuscript significantly which reduced the number of references from 268 to 164.
What is the purpose of putting together Parkinson's disease and Roux-en-Y gastric bypass?
Response: We have eliminated the section on strategies to restore DA deficits that included Roux-en-Y gastric bypass.
The role of microbiota is largely displayed and then treatment options (fecal transplantation, probiotics and weight loss surgery) are also discussed. Could these be two subjects for two different papers?
Response: Thank you for the suggestion. First, we have revised the subsections titles linking bacteria and dopaminergic system. Second, we removed the entire section #5 that included fecal microbiota transplantation, probiotics and weigh loss surgery. We’ve made the corresponding changes to the abstract, introduction and conclusion, removed the second Table and adjusted Figure 3 accordingly.
What is the purpose of introducing chemical formulas in Figures?Response:
Response: We’ve revised the figures and removed the chemical formulas.
Figure 1 has a blurry aspect.
This was corrected.
Round 2
Reviewer 2 Report
I have no further comments.